# Risk Factors for Delayed-Onset Infection after Mandibular Wisdom Tooth Extractions

**DOI:** 10.3390/healthcare11060871

**Published:** 2023-03-16

**Authors:** Ryo Miyazaki, Shintaro Sukegawa, Ken Nakagawa, Fumi Nakai, Yasuhiro Nakai, Takanori Ishihama, Minoru Miyake

**Affiliations:** 1Department of Oral and Maxillofacial Surgery, Kagawa University Faculty of Medicine, 1750-1, Ikenobe, Miki-cho, Kita-gun, Kagawa 761-0793, Japan; 2Department of Oral and Maxillofacial Surgery, Daiyukai General Hospital, 1-9-9 Sakura, Ichinomiya, Aichi 491-8551, Japan

**Keywords:** wisdom tooth extraction, infection, delay, hypertension, risk factor

## Abstract

Wisdom tooth extraction is one of the most commonly performed procedures by oral maxillofacial surgeons. Delayed-onset infection (DOI) is a rare complication of wisdom tooth extraction, and it occurs ~1–4 weeks after the extraction. In the present study, risk factors for DOI were investigated by retrospectively analyzing the cases of 1400 mandibular wisdom tooth extractions performed at Kagawa University Hospital from April 2015 to June 2022. Inclusion criteria were patients aged >15 years with a wisdom tooth extraction per our procedure. The exclusion criteria were patients with insufficient medical records, a >30-mm lesion around the wisdom tooth shown via X-ray, colonectomy, radiotherapy treatment of the mandible, the lack of panoramic images, and lesions other than a follicular cyst. The DOI incidence was 1.1% (16 cases), and univariate analyses revealed that the development of DOI was significantly associated with the Winter classification (*p* = 0.003), position (*p* = 0.003), hypertension (*p* = 0.011), and hemostatic agent use (*p* = 0.004). A multivariate logistic regression analysis demonstrated that position (OR = B for A, 7.75; *p* = 0.0163), hypertension (OR = 7.60, *p* = 0.013), and hemostatic agent use (OR = 6.87, *p* = 0.0022) were significantly associated with DOI development. Hypertension, hemostatic use, and position were found to be key factors for DOI; long-term observation may thus be necessary for patients with these risk factors.

## 1. Introduction

Wisdom tooth extraction is the most common surgery for oral surgeons, and it is necessary that surgeons minimalize the uncomfortable complications of these extractions. Potential intraoperative complications of wisdom tooth extraction are bleeding, damage to adjacent teeth, injury to surrounding tissue, displacement of teeth into adjacent spaces, fracture of the root, and maxillary tuberosity of the mandible. The common postoperative complications are pain, swelling, trismus, fever, and a dry socket, each of which can cause difficulty in chewing, speaking, and swallowing. Rare postoperative complications include postoperative infection and sensory alterations of the inferior nerve (IAN) or lingual nerve. Postoperative infection is one of the rare complications, and in the maxilla it is extremely rare [1]. However, postoperative infection occasionally occurs in the mandible, and such infections can involve abscess, pain, fever, swelling, and trismus [2,3].

Many studies have investigated frequent postoperative complications of wisdom tooth extractions such as nerve damage, a dry socket, and wound infection, but there are few reports related to delayed-onset infection (DOI) after wisdom tooth extraction. DOI is a rare complication that develops approx. 1–4 weeks after the extraction. Even though oral surgeons take precautions, such as prescribing antibiotics and advising patients about the importance of not smoking and maintaining good oral hygiene, DOIs still occur; the reported incidence of DOI has ranged from 0.5% to 1.8% [4,5,6,7]. The following factors were reported to be associated with an increased rate of total complications after wisdom tooth extraction: increased age, a positive medical history, and the position of the wisdom tooth to the inferior nerve [8]. Clinical studies have indicated the depth and the tilt of the tooth axis of the mandibular third molar as risk factors for local DOI [4], and another investigation demonstrated that the development of DOIs is related to the space distal to the second molar [9]. However, there are few reports regarding the identification of DOI risk factors from among comprehensive factors such as systemic conditions, local factors, and surgical factors related to mandibular third molar tooth extraction.

The present study was conducted to identify clinical and radiological features associated with DOI. The null hypothesis of the study was that each factor was not related to the incidence of DOI. Few studies have evaluated the multivariate relationship between clinical features and DOI, and the present study thus sought to identify DOI risk factors through performing both univariate and multivariate analyses.

## 2. Materials and Methods

### 2.1. Study Design

A retrospective clinical study of the incidence and risk factors for DOI in patients with extracted mandibular third molars at a single-center university hospital, Kagawa University Hospital, during the period from April 2015 to June 2022, was performed.

### 2.2. Ethics Statement

This study was approved by the Institutional Review Committee of the Faculty of Medicine, Kagawa University (approval no. 2022-157, approved 25 November 2022), and was conducted in full accordance with the Declaration of Helsinki. Informed consent was obtained from all patients in this study. All data were anonymized before being analyzed.

### 2.3. Patient Selection

The inclusion criteria for the patients were (1) age >15 years and (2) having undergone a wisdom tooth extraction following the described procedure. The following exclusion criteria were applied: (1) insufficient medical records, (2) a >30-mm lesion around the wisdom tooth shown via X-ray, (3) colonectomy (the removal of only crown of the tooth), (4) radiotherapy treatment of the mandible, (5) lack of panoramic images, and (6) a lesion other than a follicular cyst. With the use of these criteria, 1400 patients were enrolled in the study (Figure 1).

### 2.4. Surgical Procedure and Postoperative Management

The protocol used for managing each patient’s general condition for wisdom tooth extraction was as follows. Blood pressure was measured first before surgery and a second time after conduction and infiltration anesthesia was administered. After the surgery, the patient’s blood pressure was measured again by a specialist nurse. Oxygen saturation and pulse rate were also monitored continuously. In addition to panoramic images, corn-beam computed tomography (CBCT) images were obtained from the patients with a wisdom tooth close to the IAN.

All tooth extraction procedures were performed by residents or oral surgeons who had passed the Japanese Society of Oral and Maxillofacial Surgeons board examination for oral and maxillofacial surgery, under guidance by three experienced oral and maxillofacial surgeons (SS, FN, and MM). The surgeries were conducted with the patient under local anesthesia with 1:80,000 adrenalin with 2% lidocaine (ORA Injection Dental Cartridge, GC Showayakuhin Corp., Tokyo, Japan), with or without intravenous sedation or under general anesthesia following the patient’s preference. All surgeries were performed with sterile instruments and materials. To close the wound, 3-0 silk sutures (Alfresa Pharma Corp., Osaka, Japan) were used. Primary closure was performed whenever possible, but secondary healing was performed if this was not possible.

After the extraction, an antibiotic (amoxicillin 250 mg every 8 h for 2 days, or clarithromycin 200 mg every 12 h for 2 days for patients with penicillin allergy) and a nonsteroidal anti-inflammatory drug (loxoprofen sodium hydrate 60 mg every 6–8 h) or acetaminophen 500 mg every 6–8 h were prescribed. At ≥1 week after the extraction, the sutures were removed. At the suture removal, all patients were advised again to contact our department for any problems related to extraction, and in such cases, our consultation was conducted within a few days.

### 2.5. Outcome Variables

The patients’ clinical data were examined by three oral surgeons (RM, SS, and KN) in a review of the patients’ panoramic X-ray images on the Picture Archiving and Communication System (PACS) and past electronic medical records. DOI was defined as inflammation around the wound with purulent discharge that occurred >1 week after the extraction [8,9].

### 2.6. Predictive Variables

Attributes (sex, age)Operative variables

The following surgical variables were examined: simultaneous left and right extraction, simultaneous maxilla and mandible extraction, and the surgeon’s specialist qualification (Japanese Society of Oral and Maxillofacial Surgeons).

Anatomical variables

Wisdom tooth variables included the Winter classification, position, right or left side, the number of roots, and root canal treatment. The imaging evaluations included the use of computed tomography (CT) (CBCT and medical CT) and the imaging features of wisdom tooth lesions (follicular cyst and radicular cyst).

Physical status

The following data were obtained: height, weight, body mass index (BMI), smoking habit, alcohol consumption, hypertension, diabetes, bisphosphonate medications, corticosteroid therapy, contraceptives medications, hemostatic agent, and perioperative blood pressure. Hypertension was defined based on a physician’s diagnosis. Diabetes was defined as >6.5% HbA1c [10].

### 2.7. Statistical Analysis

In this study, data were recorded in an electronic database using Microsoft Excel. For the statistical analyses, the digital database used was JMP ver. 14.2.0 for Macintosh (SAS, Cary, NC, USA). Categorical variables were presented as numbers and percentages, while continuous variables were presented as mean and standard deviations. For the comparisons of pairs of groups, the chi-square test or Fisher’s exact test was used for categorical variables, and the Mann–Whitney U test was used for continuous variables. Adjusted odds ratios (ORs) to control the simultaneous effects of multiple covariates were obtained. Statistical significance was defined at *p* < 0.05.

## 3. Results

### 3.1. Univariate Analyses

A total of 1400 lower third molars were extracted during the study period. The incidence of DOI was 1.1% at 16 sites. Table 1 summarizes the results of the statistical analyses. The development of DOI was significantly associated with the Winter classification (*p* = 0.003), position (*p* = 0.003), hypertension (*p* = 0.011), and use of a hemostatic agent (*p* = 0.004).

### 3.2. Multivariate Logistic Regression Model Results

A multivariate logistic regression model for the occurrence of DOI was next performed. The selected items were significant variables in the bivariate analysis and variables with higher correlation coefficients (hemostatic agent, hypertension, position, Winter class), sex, and age. The results of the multivariate logistic regression analysis demonstrated that hemostatic agent use, hypertension, and position were significantly associated with the development of DOI. Position (OR = B for A, 7.75; *p* = 0.0163) and hypertension (OR = 7.60, *p* = 0.013) had high ORs for the extracted variables. The use of a hemostatic agent (OR = 6.87, *p* = 0.0022) was also significant.

## 4. Discussion

Although DOI is a rare complication of wisdom tooth extractions (which are one of the most frequent surgeries performed by oral and maxillofacial surgeons), a DOI can result in severe physical and emotional burdens. Our present retrospective analyses identified risk factors for the development of a DOI after the extraction of a wisdom tooth.

The incidence of DOI in previous investigations ranged from 1.5% to 3.7% [1,6,10,11,12,13], and our finding of a 1.1% incidence is similar to these values. In the present patient series (n = 1400), the tooth extraction procedures were performed by surgeons with different levels of experience. A surgeon’s lack of experience was reported to be a major factor associated with postoperative complications [14]. The univariate analyses conducted herein detected no significant difference in the DOI rate between the extractions performed by the residents and those performed by the specialists. A reason contributing to this result might be that all extractions were performed under the guidance of highly experienced oral and maxillofacial surgeons in our department. In addition, 3-0 silk was used as the suture instead of absorbent thread, for medical and economic reasons. Our DOI result is similar to those of previous reports; however, the difference in operators and the use of the 3-0 silk suture did not seem to affect the infection rate.

The most common age of onset for a DOI is the teens to early twenties [11,15], and DOI was reported to be the most common secondary infection in a group of patients between 12 and 24 years old [7,15].

The occurrence of DOI has been described as most frequent at 1 month post extraction [1,6,10,14]. In the present patient series, the DOIs occurred at an average of 29.1 postoperative days. Food impaction was suggested to be a risk factor for DOI [4,6]; after wound healing, it might be more difficult for food debris to escape from the socket, and this is more likely to happen at ~1 month after the surgery. It is therefore important to inform patients about the possibility of a DOI occurring several weeks after their extractions.

A younger age, total tissue coverage, deeper impaction, lower Nolla stage, mesioangular direction, and full bone coverage have been suggested as DOI risk factors [9,11], but the precise list of DOI risk factors has not been established. The results of our present univariate analyses revealed that the Winter class (*p* < 0.01), position (*p* < 0.01), hypertension (*p* < 0.01), and use of a hemostatic agent (*p* < 0.01) were significantly associated with DOI, and the multivariate logistic regression model identified hemostatic agent use, hypertension, and position as significant factors for the development of a DOI. Aspects of the patient’s physical status, such as diabetes, the use of a bisphosphonate, corticosteroid, or contraceptive, and the presence of a radicular cyst or root canal treatment were not significantly associated with DOI. These variables thus do not seem to be key factors for DOIs. Gender, the surgeon’s experience, the patient’s medical condition, smoking, and the use of an oral contraceptive have been reported to be related to postoperative complications [16] The logistic regression analysis in one of our earlier investigations demonstrated that the simultaneous extraction of left and right mandibular wisdom teeth is a risk factor for DOI [1]; the reason is thought to be that the simultaneous extraction of the mandibular wisdom tooth on both sides induces swelling and trismus and leads to an unsanitary condition in the patient’s mouth. The present study’s univariate analyses detected no significant difference in the DOI rate between the cases with simultaneous left and right extraction and those with simultaneous maxilla and mandible extraction.

The association of the wisdom tooth’s position with the development of a DOI that the present study observed herein is consistent with past reports. It is thought that the position is related to the amount of bone coverage, and that a deeper wisdom tooth needs a more extensive alveolar ostectomy, greater tooth sectioning, and a longer operation time. In addition, the restricted space causes difficulty in self-cleaning and [17]. The proper surgical technique to reduce the amount of ostectomy is thus necessary.

The present study reported that intraoperative hemostatic treatment is significantly associated with the development of infections, including DOI [1]. In our department, oxidized cellulose is available as a hemostasis agent. There are few reports about susceptibility to infection in relation to the use of oxidized cellulose, which is reported to take 2 weeks to absorb [18]. Generally, age, gender, the site of extraction, tobacco use, oral contraceptive use, anesthesia, and the surgeon’s experience are frequently cited risk factors for wisdom tooth extraction complications [8]. Possible explanations for the increased incidence of DOIs caused by hemostasis agent use could include selection bias (i.e., more difficult extraction or extractions with preoperative infection). In addition, it is hypothesized that bacteria can become attached to the remaining hemostasis agent, causing a DOI. This possibility indicates that only the smallest necessary quantity of a hemostasis agent should be used, and any excess should be removed once the hemostatic effect has been achieved.

Hypertension was highly correlated with DOI in our present analyses, whereas the patient’s perioperative blood pressure was not. Our present results provide the first clinical data to be reported regarding DOIs, and they are significant. Generally, hypertension is considered a risk factor for tooth loss due to periodontal disease [19]. It has been speculated that increased blood pressure is likely to cause both the spread of inflammation and secondary damage to the vascular endothelium [20]. These factors might affect the development of a DOI, but the exact mechanism of DOI development remains unknown. However, the identification of hypertension and hemostasis agent use as risk factors is a new discovery; new criteria and long-term observation may thus be necessary.

Antibiotics are generally prescribed to prevent postoperative infections, and patients with immunodeficiency in particular are prescribed more antibiotics [21]. Unfortunately, antibiotic resistance has become a serious public health issue worldwide [22]. Even short-duration or single amoxicillin administration causes a reduction in the number of strains that are susceptible to amoxicillin [23,24]. The optimal timing of antibiotic administration (preoperative, postoperative, or both) is not established [25,26]. The current best evidence described in a review suggests that antibiotic use reduces surgical site infections but not by enough to overcome the concerns about adverse effects and antimicrobial resistance, or to justify the routine use of antibiotics [27]. Short-term intraoral amoxicillin administration was applied in the present patients but it did not prevent the occurrence of DOI. Further research is necessary to determine the proper perioperative administration of antibiotics in wisdom tooth extractions.

The treatment for DOI is not well-defined. An oral antibiotic is commonly administered for a DOI. *Fusobacterium*, *Prevotella*, *Bacteroides*, and *Peptostreptococcus* have been identified in DOIs, and the antibiotic clindamycin has been the most effective for DOIs, followed by metronidazole and amoxicillin/clavulanate. Amoxicillin alone is not sufficiently effective for *Fusobacterium* or *Prevotella* [28]. When antibiotic treatment is not successful, surgical debridement of the extraction site is recommended [29]. Removal of the granulation tissue from the socket, debridement of bone particles, and removal of any foreign matter are thought to be essential for DOI treatment [29].

In this study, antibiotic treatment was performed in all cases. For the patients with a mild DOI, amoxicillin or sitafloxacin was used. For the patients with a severe DOI, ceftriaxone or sulbuctam/ampicillin was administered intravenously. Four patients underwent surgical debridement. After the treatment, all 16 of the cases of DOI healed well. As in previous reports, the use of an antibiotic and then a surgical procedure, if necessary, seem to be the most suitable treatments for DOIs.

There are some study limitations to consider. The patient population was retrospectively drawn from a single hospital. There was a bias in the degree of difficulty of the tooth extraction, which may have affected the surgical method selected by the oral surgeons. Even though in the present study all surgeons followed our surgical protocol to standardize the surgical procedures, the levels of experiences among the providers were different. Besides, CBCT was used for not all the cases. We would like to conduct further research through prospective studies. Secondly, although another investigation indicated that the incidence of infection was not significantly different between cases with secondary closure versus primary closure [30], our suture protocol was not established. In addition, whether the patients with DOIs came back to our department after their sutures were removed depended on the patients and their symptoms. It is thus necessary to take this uncertainty into account in future studies.

## 5. Conclusions

The results of this retrospective study of 1400 cases demonstrated that hypertension, the position of the wisdom tooth, and the use of a hemostasis agent were significantly associated with the development of a DOI. To our best knowledge, the present study is the first to report that the presence of hypertension affects the incidence of DOI. Especially for patients with any of these three factors, long-term observation and professional oral care might be important after wisdom tooth extraction.

## Figures and Tables

**Figure 1 healthcare-11-00871-f001:**
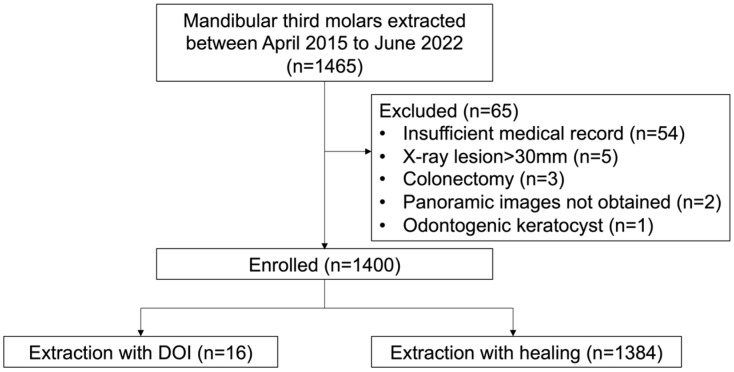
Flowchart of the patient enrollment based on the inclusion and exclusion criteria.

**Table 1 healthcare-11-00871-t001:** Univariate analyses of predictive variables related to delayed-onset infections after mandibular wisdom tooth extractions.

		Healing	DOI	*p*-Value
Outcome (%)		1384 (98.9%)	16 (1.1%)	
** *Attribute variables* **				
Age		32.4 ± 14.1	37.7 ± 20.9	0.143
Sex	Male	564	5	0.610
	Female	820	11	
** *Operative variables* **				
Simultaneous left and right extraction	Yes	440	3	0.417
	No	944	13	
Simultaneous maxilla and mandible extraction	Yes	344	6	0.251
	No	1040	10	
Specialist	Yes	175	4	0.137
	No	1209	12	
** *Anatomical variables* **				
Class	Ⅰ	482	3	0.003
	Ⅱ	712	6	
	Ⅲ	190	7	
Position	A	758	2	0.003
	B	487	10	
	C	139	4	
Right or left	Left	706	5	0.136
	Right	678	11	
No. of wisdom tooth roots	Immature root	100	4	0.058
	1	492	4	
	2	778	8	
	3	14	0	
Wisdom tooth follicular cyst	Yes	44	2	0.095
	No	1340	14	
Wisdom tooth radicular cyst	Yes	8	0	1.000
	No	1376	16	
Wisdom tooth root canal filling	Yes	7	0	1.000
	No	1377	16	
** *Physical status* **				
Height		162.2 ± 8.5	160.1 ± 9.7	0.319
Weight		58.6 ± 12.3	59.9 ± 19.2	0.672
BMI		22.1 ± 3.7	23.0 ± 4.6	0.343
Hypertension	Yes	76	4	0.011
	No	1308	12	
Diabetes	Yes	34	0	1.000
	No	1350	16	
Bisphosphonate	Yes	10	1	0.119
	No	1372	15	
Corticosteroid	Yes	24	0	1.000
	No	1360	16	
Contraceptives	Yes	11	0	1.000
	No	1373	16	
Smoking	Yes	179	3	0.452
	No	1205	13	
Alcohol consumption	Yes	212	3	0.725
	No	1172	13	
Hemostatic agent	Yes	58	4	0.004
	No	1326	12	
Preoperative SBP		125.3 ± 17.0	133 ± 20.8	0.075
Preoperative DBP		77.1 ± 24.3	77.0 ± 13.9	0.984
SBP after local anesthesia		121.6 ± 18.3	121.4 ± 19.9	0.955
DBP after local anesthesia		71.0 ± 13.9	71.9 ± 15.0	0.796
Postoperative SBP		122.1 ± 17.4	121.8 ± 14.6	0.946
Postoperative DBP		73.3 ± 14.6	70.0 ± 13.9	0.361

BMI: body mass index, DBP: diastolic blood pressure, SBP: systolic blood pressure.

## Data Availability

The data obtained in this study are unavailable due to privacy or ethical restrictions.

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
