# Peer review of "Risk Factors for Delayed-Onset Infection after Mandibular Wisdom Tooth Extractions"

_healthcare, 2023, doi:10.3390/healthcare11060871_

Round 1

Reviewer 1 Report

- Who performed the extractions? Oral surgeons, residents, dentists or students? Do you think it would affect the final outcome and your findings? 

- What was the reason behind the decision whether you administered or not IV sedation and general anesthesia in the extraction procedures? 

- Why have you used 3-0 silk suture and not a 4-0 vicryl or a 4-0 chromic gut suture? Do you think it played any role? 

- Did you have an established protocol when you have removed the sutures? Do you think it would make a difference if you have used a resorbable suture instead? Did you have primary closure after your suturing or the sites healed by secondary intention? Was that part of your protocol? Was your protocol standardized? 

- Avoid phrases such as "Generally speaking..." line 165

- Why have you chosen in your inclusion criteria pts. to have underwent a wisdom tooth extraction? 

- Do you think the physical status of the pts. played any role to your final results (e.g. diabetes, steroids etc)? 

Do you think the presence of a radicular cyst/root canal treatment played any role to your final results? 

- Explain/discuss the results from the lines 128-131.

- Do you think your study would lead to some important recommendations in order to help the readers and eliminate this complication in the future? 

- The Discussion section is poor and needs more literature to support/justify/discuss your results.

Author Response

March 4, 2023

Reviewer 1

We would like to resubmit our manuscript entitled “Risk factors for delayed-onset infection after mandibular wisdom tooth extractions” to Healthcare.

We very much appreciate the valuable comments of the reviewers; we have carefully revised the manuscript to address all their concerns and responded to each of the comments in a point-by-point fashion below.

  1. Who performed the extractions? Oral surgeons, residents, dentists or students? Do you think it would affect the final outcome and your findings?

We added “the tooth extraction procedures were performed by surgeons with different levels of experience. A surgeon's lack of experience was reported to be a major factor associated with postoperative complications [14]. The univariate analysis conducted herein detected no significant difference in the DOI rate between the extractions performed by the residents and those performed by the specialists. A reason contributing to this result might be that all extractions were performed under the guidance of highly experienced oral and maxillofacial surgeons in our department.”(line 180-187)

  1. What was the reason behind the decision whether you administered or not IV sedation and general anesthesia in the extraction procedures? 

We added “The surgeries were conducted with the patient under local anesthesia with 1:80,000 adrenalin with 2% lidocaine (ORA Injection Dental Cartridge, GC Showayakuhin Corp., Tokyo), with or without intravenous sedation or under general anesthesia following the patient's wishes”(line 98-101 ).

  1. Why have you used 3-0 silk suture and not a 4-0 vicryl or a 4-0 chromic gut suture? Do you think it played any role? 

We added “In addition, 3-0 silk was used as the suture instead of absorbent thread, for medical economic reasons. Our DOI result is similar to those of previous reports, however, and the difference in operators and the use of 3-0 silk suture did not seem to affect the infection rate.”(line 187-190)

  1. Did you have an established protocol when you have removed the sutures? Do you think it would make a difference if you have used a resorbable suture instead? Did you have primary closure after your suturing or the sites healed by secondary intention? Was that part of your protocol? Was your protocol standardized? 

We added “Primary closure was performed whenever possible, but secondary healing was performed if this was not possible.” (Page 3, line103-105), “In addition, 3-0 silk was used as the suture instead of absorbent thread, for medical economic reasons. Our DOI result is similar to those of previous reports, however, and the difference in operators and the use of 3-0 silk suture did not seem to affect the infection rate.”(Page 7, line 187-190), and “Secondly, although another investigation indicated that the incidence of infection was not significantly different between cases with secondary closure versus primary closure [30], our suture protocol was not established.”(,line 287-289)

  1. Avoid phrases such as "Generally speaking..." line 165

We changed “Generally speaking” to “ Generally”.

  1. Why have you chosen in your inclusion criteria pts. to have underwent a wisdom tooth extraction? 

We added “The inclusion criteria for the patients were (1) age >15 years, and (2) having undergone a wisdom tooth extraction following the described procedure. The following exclusion criteria were applied: (1) insufficient medical records, (2) a >30-mm lesion around the wisdom tooth shown by X-ray, (3) colonectomy, (4) radiotherapy treatment of the mandible, (5) lack of panoramic images, and (6) a lesion other than a follicular cyst. With the use of these criteria, 1,400 patients were enrolled in the study (Fig. 1).”(Page 2, line 78-84) and “The patient population was retrospectively drawn from a single hospital. There was a bias in the degree of difficulty of tooth extraction, which may have affected the surgical method selected by the oral surgeons.”(line 283-287).

  1. Do you think the physical status of the pts. played any role to your final results (e.g. diabetes, steroids etc)?

 We added “Aspects of the patient's physical status such as diabetes, the use of a bisphosphonate, corticosteroid, or contraceptive, and the presence of a radicular cyst or root canal treatment were not significantly associated with DOI. These variables thus do not seem to be key factors for DOIs.”(line 208-210).

  1. Do you think the presence of a radicular cyst/root canal treatment played any role to your final results? 

We added “Aspects of the patient's physical status such as diabetes, the use of a bisphosphonate, corticosteroid, or contraceptive, and the presence of a radicular cyst or root canal treatment were not significantly associated with DOI. These variables thus do not seem to be key factors for DOIs.”(line 208-210).

  1. Explain/discuss the results from the lines 128-131.

We added “The association of the wisdom tooth's position with the development of a DOI that we observed herein is consistent with past reports. It is thought that the position is related to the amount of bone coverage, and that a deeper wisdom tooth needs a more extensive alveolar bone osteotomy, greater tooth sectioning, and a longer operation time. In addition, the restricted space causes the difficulty in self-cleaning and [17]. The proper surgical technique to reduce the amount of bone osteotomy is thus necessary.

We reported that intraoperative hemostatic treatment is significantly associated with the development of infections including DOI [1]. In our department, oxidized cellulose is available as a hemostasis agent. There are few reports about susceptibility to infection in relation to the use of oxidized cellulose, which is reported to takes 2 weeks to absorb[18]. Generally, age, gender, the site of extraction, tobacco use, oral contraceptive use, anesthesia, and the surgeon's experience are frequently cited risk factors for wisdom tooth extraction complications [8]. Possible explanations for the increased incidence of DOIs caused by hemostasis agent use could include selection bias (i.e., more difficult extraction or extractions with preoperative infection). In addition, it is hypothesized that bacteria can become attached to the remaining hemostasis agent, causing a DOI. This possibility indicates that only the smallest necessary quantity of a hemostasis agent should be used, and any excess should be removed once the hemostatic effect has been achieved.

Hypertension was highly correlated with DOI in our present analyses, whereas the patient's perioperative blood pressure was not. Our present results provide the first clinical data to be reported regarding DOIs, and they are significant. Generally, hypertension is considered a risk factor for tooth loss due to periodontal disease [19]. It has been speculated that increased blood pressure is likely to cause both the spread of inflammation and secondary damage to the vascular endothelium [20]. These factors might affect the development of a DOI, but the exact mechanism of DOI development remains unknown. However, the identification of hypertension and hemostasis agent use as risk factors is a new discovery; new criteria and long-term observation may thus be necessary. (line 222 to 251).

  1. Do you think your study would lead to some important recommendations in order to help the readers and eliminate this complication in the future?

We added “Our study is the first to report that the presence of hypertension affects the incidence of DOI. Especially for patients with any of these three factors, long-term observation and professional oral care might be important after wisdom tooth extraction.”(line 298-300).

  1. The Discussion section is poor and needs more literature to support/justify/discuss your results.

 We added more explanation in the Discussion section following your kind advice.

This manuscript has not been published or is not under consideration for publication elsewhere. All authors have read the manuscript and have concurred with this submission.

Thank you again for your assistance. We look forward to hearing from you in due course.

Shintaro Sukegawa, DDS, Ph.D.

Associate Professor

Department of Oral and Maxillofacial Surgery

Faculty of Medicine, Kagawa University

1750-1 Ikenobe, Miki-cho, Kita-gun, Kagawa 761-0793, Japan

Phone No: +81-87-891-2227

Fax No: +81-87-891-2228

E-mail address: sukegawa.shintaro.u7@med.kagawa-u.ac.jp

Reviewer 2 Report

The study is a retrospective study that aims to investigate the risk factors for delayed-onset infection (DOI) after mandibular wisdom tooth extractions. The study is interesting. There are some issues that should be considered;

The abstract is too short. It is recommended to improve it by 1) describing the inclusion and exclusion criteria, 2) better description of the results with the p values of the statistical results, and 3) better description of the conclusions.

The introduction is too short. It is recommended to improve it by allonging it and demonstrating the cause of conducting this study and the gap in the literature that will be addressed by conducting this study. In addition at the end of the introduction, a paragraph should be added in which the authors describe in detail the aim of the study.

In the methods section, the authors described that the surgical extractions were conducted by three professional oral and maxillofacial surgeons. And then in the results section, they considered that the presence of the specialist was an operative variable. So it is recommended in the methods section to specify the level of each oral surgeon as it appears as a variable with results.

In discussion, it is recommended to describe in two or more paragraphs the treatment and the management of this complication.

Also in the discussion, the authors should describe the prevention strategy for this complication based on the literature and their experience.

In addition, adding a paragraph at the end of the discussion in which the authors should demonstrate the study limitations according to their experience for helping the readers better interpretation of the results of this study.

The conclusion is too short. It should be demonstrated what the authors observed based on this experience that should be addressed in future research or probably the prevention strategy of this complication based on these results.

The references are a little old. If it is possible, it would be better to increase the number of recent studies.

Author Response

March 4, 2023

Reviewer 2

We would like to resubmit our manuscript entitled “Risk factors for delayed-onset infection after mandibular wisdom tooth extractions” to Healthcare.

We very much appreciate the valuable comments of the reviewers; we have carefully revised the manuscript to address all their concerns and responded to each of the comments in a point-by-point fashion below.

  1. The abstract is too short. It is recommended to improve it by 1) describing the inclusion and exclusion criteria, 2) better description of the results with the p values of the statistical results, and 3) better description of the conclusions.

We added “Inclusion criteria were patients aged >15 years with a wisdom tooth extraction per our procedure. The exclusion criteria were patients with insufficient medical records, a >30-mm lesion around the wisdom tooth shown by X-ray, colonectomy, radiotherapy treatment of the mandible, the lack of panoramic images, and lesion other than a follicular cyst. The DOI incidence was 1.1% (16 cases), and a univariate analysis revealed that the development of DOI was significantly associated with the Winter classification (p<0.01), position (p<0.01), hypertension (p<0.01), and hemostatic agent use (p<0.01). A multivariate logistic regression analysis demonstrated that position (OR=B for A, 7.75; p=0.0163), hypertension (OR=7.60, p=0.013), and hemostatic agent use (OR=6.87, p=0.0022) were significantly associated with DOI development. Hypertension was found to be a key factor for DOI; long-term observation may thus be necessary for patients with hypertension.” (line 18 to 27)

  1. The introduction is too short. It is recommended to improve it by allonging it and demonstrating the cause of conducting this study and the gap in the literature that will be addressed by conducting this study. In addition at the end of the introduction, a paragraph should be added in which the authors describe in detail the aim of the study.

 We added more background and explanation about this research following your kind advice. Besides, we added “We thus conducted the present study to identify clinical and radiological features associated with DOI. The null hypothesis of the study was that each factor was not related to the incidence of DOI. Few studies have evaluated the multivariate relationship between clinical features and DOI, and we thus sought to identify DOI risk factors by performing both univariate and multivariate analyses.”(line 59 to 63).

  1. In the methods section, the authors described that the surgical extractions were conducted by three professional oral and maxillofacial surgeons. And then in the results section, they considered that the presence of the specialist was an operative variable. So it is recommended in the methods section to specify the level of each oral surgeon as it appears as a variable with results.

 We added “All tooth extraction procedures were performed by residents or oral surgeons who had passed the Japanese Society of Oral and Maxillofacial Surgeons board examination for oral and maxillofacial surgery, under guidance by three experienced oral and maxillofacial surgeons (SS, FN, and MM).”(line 95 to 98)

  1. In discussion, it is recommended to describe in two or more paragraphs the treatment and the management of this complication.

 We added more explanation about the treatment and the management of this complication. (line 253 to 274)

  1. Also in the discussion, the authors should describe the prevention strategy for this complication based on the literature and their experience.

 We added “the identification of hypertension and hemostasis agent use as risk factors is a new discovery; new criteria and long-term observation may thus be necessary.”( line 249 to 251)

  1. In addition, adding a paragraph at the end of the discussion in which the authors should demonstrate the study limitations according to their experience for helping the readers better interpretation of the results of this study.

 We added “There are some study limitations to consider. The patient population was retrospectively drawn from a single hospital. There was a bias in the degree of difficulty of tooth extraction, which may have affected the surgical method selected by the oral surgeons. We would like to conduct further research through prospective studies. Secondly, although another investigation indicated that the incidence of infection was not significantly different between cases with secondary closure versus primary closure [30], our suture protocol was not established. In addition, whether the patients with DOIs came back to our department after their sutures were removed depended on the patients and their symptoms. It is thus necessary to take this uncertainty into account in future studies.

”(line 283 to 292)

  1. The conclusion is too short. It should be demonstrated what the authors observed based on this experience that should be addressed in future research or probably the prevention strategy of this complication based on these results.

 We added more details about this study and our strategy to prevent DOI.(line 295 to 300)

  1. The references are a little old. If it is possible, it would be better to increase the number of recent studies.

 We added more recent references.

This manuscript has not been published or is not under consideration for publication elsewhere. All authors have read the manuscript and have concurred with this submission.

Thank you again for your assistance. We look forward to hearing from you in due course.

Shintaro Sukegawa, DDS, Ph.D.

Associate Professor

Department of Oral and Maxillofacial Surgery

Faculty of Medicine, Kagawa University

1750-1 Ikenobe, Miki-cho, Kita-gun, Kagawa 761-0793, Japan

Phone No: +81-87-891-2227

Fax No: +81-87-891-2228

E-mail address: sukegawa.shintaro.u7@med.kagawa-u.ac.jp

Reviewer 3 Report

The study seems interesting and genuine, however the authors should address the following points to improve the quality of the manuscript:

- The abstract is too short. Please expand it within word limit (check authors' guidelines).

- The introduction is also too short. Please expand it and add more literature review, current research gap and question, clear objectives and null hypotheses.

- "DOI was defined as inflammation around the wound with 72 purulent discharge that occurred >1 week after the extraction", Please add citation for this evaluation criteria.

- Please enlist inclusion and exclusion criteria for clarity.

- The manuscript should be edited using passive voice writing.

- Please add study limitations to discussion.

- Please expand the conclusion section to cover the study outcomes and reflect the significance of the study.

Author Response

March 4, 2023

Reviewer 3

We would like to resubmit our manuscript entitled “Risk factors for delayed-onset infection after mandibular wisdom tooth extractions” to Healthcare.

We very much appreciate the valuable comments of the reviewers; we have carefully revised the manuscript to address all their concerns and responded to each of the comments in a point-by-point fashion below.

  1.  The abstract is too short. Please expand it within word limit (check authors' guidelines).

We added “Inclusion criteria were patients aged >15 years with a wisdom tooth extraction per our procedure. The exclusion criteria were patients with insufficient medical records, a >30-mm lesion around the wisdom tooth shown by X-ray, colonectomy, radiotherapy treatment of the mandible, the lack of panoramic images, and lesion other than a follicular cyst. The DOI incidence was 1.1% (16 cases), and a univariate analysis revealed that the development of DOI was significantly associated with the Winter classification (p<0.01), position (p<0.01), hypertension (p<0.01), and hemostatic agent use (p<0.01). A multivariate logistic regression analysis demonstrated that position (OR=B for A, 7.75; p=0.0163), hypertension (OR=7.60, p=0.013), and hemostatic agent use (OR=6.87, p=0.0022) were significantly associated with DOI development. Hypertension was found to be a key factor for DOI; long-term observation may thus be necessary for patients with hypertension.” (line 18 to 27)

  1. The introduction is also too short. Please expand it and add more literature review, current research gap and question, clear objectives and null hypotheses.

 We added more background and explanation about this research following your kind advice. Besides, we added “The null hypothesis of the study was that each factor was not related to the incidence of DOI. Few studies have evaluated the multivariate relationship between clinical features and DOI, and we thus sought to identify DOI risk factors by performing both univariate and multivariate analyses.”(line 60 to 63).

  1. "DOI was defined as inflammation around the wound with 72 purulent discharge that occurred >1 week after the extraction", Please add citation for this evaluation criteria.

 We added “DOI was defined as inflammation around the wound with purulent discharge that occurred >1 week after the extraction [8]  [9].”(line 117 to 118)

  1. Please enlist inclusion and exclusion criteria for clarity.

 We added “The inclusion criteria for the patients were (1) age >15 years, and (2) having undergone a wisdom tooth extraction following the described procedure. The following exclusion criteria were applied: (1) insufficient medical records, (2) a >30-mm lesion around the wisdom tooth shown by X-ray, (3) colonectomy, (4) radiotherapy treatment of the mandible, (5) lack of panoramic images, and (6) a lesion other than a follicular cyst.” (line 78 to 83)

  1. The manuscript should be edited using passive voice writing.

 We changed our manuscript to passive voice writing.

  1. Please add study limitations to discussion

 We added “There are some study limitations to consider. The patient population was retrospectively drawn from a single hospital. There was a bias in the degree of difficulty of tooth extraction, which may have affected the surgical method selected by the oral surgeons. We would like to conduct further research through prospective studies. Secondly, although another investigation indicated that the incidence of infection was not significantly different between cases with secondary closure versus primary closure [30], our suture protocol was not established. In addition, whether the patients with DOIs came back to our department after their sutures were removed depended on the patients and their symptoms. It is thus necessary to take this uncertainty into account in future studies.

”(line 283 to 292)

  1. Please expand the conclusion section to cover the study outcomes and reflect the significance of the study.

 We added more details about this study and our strategy to prevent DOI.(line 295 to 300)

This manuscript has not been published or is not under consideration for publication elsewhere. All authors have read the manuscript and have concurred with this submission.

Thank you again for your assistance. We look forward to hearing from you in due course.

Shintaro Sukegawa, DDS, Ph.D.

Associate Professor

Department of Oral and Maxillofacial Surgery

Faculty of Medicine, Kagawa University

1750-1 Ikenobe, Miki-cho, Kita-gun, Kagawa 761-0793, Japan

Phone No: +81-87-891-2227

Fax No: +81-87-891-2228

E-mail address: sukegawa.shintaro.u7@med.kagawa-u.ac.jp

Round 2

Reviewer 1 Report

- Please replace "bone osteotomy", with ostectomy or removal of bone. 

- In your inclusion criteria, why was important that the patients had to previously undergo a wisdom tooth extraction following the described procedure? New pts. were not accepted in your study? 

- Please change in the manuscript the phrases "we" and "our study" with "in the present study". 

- What do you mean by "colonectomy"? (line 83)

- Have you used only panoramic images for your third molar extractions? Even when a third molar was close to the IAN? How about the use of a CBCT in these cases? 

- Why the use of 750 mg Amoxicillin and not 500 mg? Do you have any evidence to support that selection? 

- Please replace in the mauscript the word "drugs" with "medications"

-  Please replace "patient wishes" with "patient preference"

- Univariate "analyses", not "analysis"

- Add a "limitations" section at your "Discussion" section stating all the different limitations of your study, such as i. the different level of experience among the providers, ii. the lack of a CBCT etc.

- The lines 313-316 belong to the Discussion section only. 

- Moderate English changes is required. 

Author Response

March 11, 2023

Reviewer 1

We would like to resubmit our manuscript entitled “Risk factors for delayed-onset infection after mandibular wisdom tooth extractions” to Healthcare.

We very much appreciate the valuable comments of the reviewers; we have carefully revised the manuscript to address all their concerns and responded to each of the comments in a point-by-point fashion below.

  1. Please replace "bone osteotomy", with ostectomy or removal of bone. 

We replaced "bone osteotomy" with ostectomy”

  1. In your inclusion criteria, why was important that the patients had to previously undergo a wisdom tooth extraction following the described procedure? New pts. were not accepted in your study?

We added the explanation “Even though in the present study, all surgeons followed our surgical protocol to standardize the surgical procedures, the levels of experiences among the providers were different.”

  1. Please change in the manuscript the phrases "we" and "our study" with "in the present study". 

We changed in the manuscript the phrases "we" and "our study" with "in the present study". 

  1.  What do you mean by "colonectomy"? (line 83)

We added “colonectomy (the removal of only crown of the tooth)”(line 86-87)

  1. Have you used only panoramic images for your third molar extractions? Even when a third molar was close to the IAN? How about the use of a CBCT in these cases? 

We added “In addition to panoramic images, corn-beam computed tomography (CBCT) images were obtained from the patients with a wisdom tooth close to the IAN. ”(line 105-107)

  1. Why the use of 750 mg Amoxicillin and not 500 mg? Do you have any evidence to support that selection? 

We are really sorry that we should have written amoxicillin 250 mg every 8 hr for 2 days.

We collected this description.(line 119-120)

  1. Please replace in the mauscript the word "drugs" with "medications" 

We replaced the word "drugs" with "medications".

  1. Please replace "patient wishes" with "patient preference"

 We replaced "patient wishes" with "patient preference".

  1. Univariate "analyses", not "analysis"

We replaced "analysis" with "analyses".

  1. Add a "limitations" section at your "Discussion" section stating all the different limitations of your study, such as i. the different level of experience among the providers, ii. the lack of a CBCT etc.

We added “Even though in the present study, all surgeons followed our surgical protocol to standardize the surgical procedures, the levels of experiences among the providers were different. Besides, CBCT was used for not all the cases.(line 222 to 251).

  1. The lines 313-316 belong to the Discussion section only. 

Thank you for your valuable comment. The declaration part has been modified according to the posting rules.

  1. Moderate English changes is required. 

We modified our manuscript following your kind review.

This manuscript has not been published or is not under consideration for publication elsewhere. All authors have read the manuscript and have concurred with this submission.

Thank you again for your assistance. We look forward to hearing from you in due course.

Shintaro Sukegawa, DDS, Ph.D.

Associate Professor

Department of Oral and Maxillofacial Surgery

Faculty of Medicine, Kagawa University

1750-1 Ikenobe, Miki-cho, Kita-gun, Kagawa 761-0793, Japan

Phone No: +81-87-891-2227

Fax No: +81-87-891-2228

E-mail address: sukegawa.shintaro.u7@med.kagawa-u.ac.jp

Reviewer 2 Report

The study investigates an interesting topic. These are the noticed issues after revision.

In the abstract, the authors should report the exact p-value of each variable rather than “p<0.01”.

The conclusion in the abstract is not complete, a revision of it should be performed and should be improved.

The description of the considered variables in the methods section is still somehow confusing with a lack of order. A careful revision is needed.

Some abbreviations were not defined at first mention in the manuscript, such as PACS. A careful revision of this point is needed.

The same point for the p-value is present in the results section. The exact p-value should be also provided in the text.

It has been noticed that the authors used many times “we” and “our”. It is preferable to avoid using them as possible as you can.

In the conclusion, the authors stated: “Our study is the first to report”. This statement should be supported by a reference of a recent systematic review that supports this conclusion. If there is no, please modify it and decrease the level of affirmation. Please carefully revise the whole manuscript for this point.

Author Response

March 11, 2023

Reviewer 2

We would like to resubmit our manuscript entitled “Risk factors for delayed-onset infection after mandibular wisdom tooth extractions” to Healthcare.

We very much appreciate the valuable comments of the reviewers; we have carefully revised the manuscript to address all their concerns and responded to each of the comments in a point-by-point fashion below.

1.In the abstract, the authors should report the exact p-value of each variable rather than “p<0.01”.

We reported the exact p-value of each variable

  1. The conclusion in the abstract is not complete, a revision of it should be performed and should be improved. 

We changed previous manuscript into " Hypertension, hemostatic use and position were found to be a key factor for DOI; long-term observation may thus be necessary for patients with these risk factors ."

3.The description of the considered variables in the methods section is still somehow confusing with a lack of order. A careful revision is needed.

We modified the description of the considered variables in the methods section following Table.1.

  1. Some abbreviations were not defined at first mention in the manuscript, such as PACS. A careful revision of this point is needed. 

We added “Picture Archiving and Communication System (PACS)”.(line134-135)

  1. The same point for the p-value is present in the results section. The exact p-value should be also provided in the text. 

We provided The exact p-value in the results section.

  1. It has been noticed that the authors used many times “we” and “our”. It is preferable to avoid using them as possible as you can.

We rewrote the manuscript as a passive style. 

  1. In the conclusion, the authors stated: “Our study is the first to report”. This statement should be supported by a reference of a recent systematic review that supports this conclusion. If there is no, please modify it and decrease the level of affirmation. Please carefully revise the whole manuscript for this point.

We added “To our best knowledge ”.(line 353)

This manuscript has not been published or is not under consideration for publication elsewhere. All authors have read the manuscript and have concurred with this submission.

Thank you again for your assistance. We look forward to hearing from you in due course.

Shintaro Sukegawa, DDS, Ph.D.

Associate Professor

Department of Oral and Maxillofacial Surgery

Faculty of Medicine, Kagawa University

1750-1 Ikenobe, Miki-cho, Kita-gun, Kagawa 761-0793, Japan

Phone No: +81-87-891-2227

Fax No: +81-87-891-2228

E-mail address: sukegawa.shintaro.u7@med.kagawa-u.ac.jp

Reviewer 3 Report

All comments were addressed and the manuscript can be accepted in the current form.

Author Response

All comments were addressed and the manuscript can be accepted in the current form.

Author response: Thank you so much for your very pleasant evaluation comment for our research in terms of study of historical evaluation in the risk factors for delayed-onset infection after mandibular wisdom tooth extractions. Thank you for your dedicated review contribution.